# Visibility as a Key Dimension to Better Health-Related Quality of Life and Mental Health: Results of the European Union Funded “ME-WE” Online Survey Study on Adolescent Young Carers in Switzerland

**DOI:** 10.3390/ijerph20053963

**Published:** 2023-02-23

**Authors:** Elena Guggiari, Marianne Fatton, Saul Becker, Feylyn Lewis, Giulia Casu, Renske Hoefman, Elizabeth Hanson, Sara Santini, Licia Boccaletti, Henk Herman Nap, Valentina Hlebec, Alexandra Wirth, Agnes Leu

**Affiliations:** 1Careum School of Health, Kalaidos University of Applied Sciences, Gloriastrasse 18a, 8006 Zurich, Switzerland; 2Careum, Pestalozzistrasse 3, 8032 Zurich, Switzerland; 3Medical Faculty, Institute for Biomedical Ethics, University of Basel, Bernoullistrasse 28, 4056 Basel, Switzerland; 4Faculty of Health and Education, Manchester Metropolitan University, Manchester M15 6BX, UK; 5School of Nursing, Vanderbilt University, Godchaux Hall 179, 461 21st Ave S, Nashville, TN 37240, USA; 6Department of Psychology, University of Bologna, Viale Berti Pichat 5, 40127 Bologna, Italy; 7The Netherlands Institute for Social Research (SCP), Postbus 16164, 2500 BD The Hague, The Netherlands; 8Department of Health and Caring Sciences, Linnaeus University, 39182 Kalmar, Sweden; 9The Swedish Family Care Competence Centre (NKA), 39232 Kalmar, Sweden; 10Centre for Socio-Economic Research on Aging, IRCCS INRCA-National Institute of Health and Science on Aging, Via Santa Margherita 5, 60124 Ancona, Italy; 11Anziani e Non Solo Società Cooperativa Sociale, 41012 Carpi, Italy; 12Vilans-The National Centre of Expertise for Long-Term Care in The Netherlands, Churchilllaan 11, 3527 GV Utrecht, The Netherlands; 13Faculty of Social Sciences, University of Ljubljana, Kardeljeva pl. 5, 1000 Ljubljana, Slovenia

**Keywords:** adolescent young carers (AYCs), mental health, well-being, AYCs characteristics, support, visibility, HRQL

## Abstract

This paper examines the health-related quality of life (HRQL) and mental health of adolescent young carers (AYCs) aged 15–17 in Switzerland, based on data collected within the Horizon 2020 project ‘Psychosocial support for promoting mental health and well-being among AYCs in Europe’ (ME-WE). It addresses the following questions: (1) Which characteristics of AYCs are associated with lower HRQL and with higher level of mental health problems? (2) Do AYCs who are less visible and less supported report a lower HRQL and more mental health issues than other AYCs? A total of 2343 young people in Switzerland, amongst them 240 AYCs, completed an online survey. The results show that female AYCs and AYCs with Swiss nationality more often reported having mental health issues than their male and non-Swiss counterparts. Furthermore, the findings show a significant association between receiving support for themselves and visibility from their school or employer and the HRQL. Moreover, AYCs who reported that their school or employer knew about the situation also reported fewer mental health issues. These findings can inform recommendations for policy and practice to develop measures aimed at raising the visibility of AYCs, which is the first step for planning AYC tailored support.

## 1. Introduction

When a family member or a person close to them has a physical or mental health issue, children or adolescents are often involved in caring activities. Across the world, many young people carry out a significant role in caring for family members or other close persons. These young people are defined in the literature as ‘young carers’ (YCs), namely, “Children and young persons under 18 who provide, or intend to provide care, assistance or support to another family member. They carry out, often on a regular basis, significant or substantial caring tasks and assume a level of responsibility which would usually be associated with an adult. The person receiving care is often a parent but can be a sibling, grandparent or other relative who is disabled, has some chronic illness, mental health problem or other condition connected with a need for care, support or supervision.” [1]. YCs can also provide care for a close person who is not a relative [2,3]. 

YCs may take over a variety of different tasks including (but not limited to) domestic tasks, intimate and personal care, emotional care and support, household management, and care of younger siblings [4]. In addition, the caring also comprises the time spent thinking about the person with care needs [5]. In other words: caring tasks can be viewed in terms of a continuum ranging from caring about someone to caring for someone [6]. There are different reasons why children, adolescents, and young people care for a relative or a close person and are therefore in a role as a ‘young carer’. One reason can be a sudden onset of illness or disability of a related person, which necessitates a child taking on caring tasks, particularly if there is no adult available or willing to provide care. The lack of affordable formal paid carers can also draw a child into a caring role [4]. Other caring situations happen gradually as a consequence of long-term conditions, which might become more severe over time [7]. Another reason for being driven into a caring role might be religious and/or cultural expectations [8,9]. Age also matters: as YCs become older and more experienced and competent, some families have growing expectations and place demands on them to provide more care [10]. 

The caring role may have negative but also positive effects on children and young people. Concerning the possible negative consequences of the caring role, YCs might experience restricted opportunities for social networking and developing peer friendship as well as taking part in leisure activities, and they have a higher risk of being bullied than their peers without a caring role [11,12]. Being a YC can also have a negative impact on their physical and mental health. They have, for instance, generally higher levels of depressive and anxiety symptoms [13] and they might feel invisible and abandoned [14]. In the case of specific illnesses, there might be an additional “stigma by association” (i.e., experiencing public disapproval as a consequence of associating with stigmatized persons), particularly if their parents have mental health problems [12], misuse of alcohol or drugs, or have HIV/AIDS [15]. Another highlighted negative consequence of caring roles concerns the education, employment, and educational opportunities of YCs. The consequences of the caring role in the education of young carers might include absenteeism and higher dropout rates from schooling, a lack of concentration in school, and lower performances [16,17,18,19]. However, the caring role might also have positive outcomes including the development of children’s knowledge, understanding, sense of responsibility, and maturity as well as a range of life, social, and transferable care-related skills [20] and a higher level of independence compared to peers without a caring role [21]. Furthermore, by taking on a caring role, children can feel ‘included’ and have an enhanced relationship with the cared-for person [12]. However, to be able to develop resilience and benefit from the possible positive consequences of the caring role, appropriate support is valuable for YCs [15,22]. In addition, previous studies have highlighted the dilemma of whether children should be protected from an inappropriate caring role in order to be able to attain their rights or if this should be considered part of family life [23,24]. Indeed, sometimes, caring can be inappropriate for YCs, and in these cases, it helps if another person or formal health care services can take over the caring tasks. 

The early identification of YCs is vital to provide appropriate support to YCs and to their family. This is strongly linked to the level of awareness of professionals and it may pose important challenges. In some cases, YCs do not want to draw attention to themselves related to a caring role [23]. Furthermore, for professionals, the situation experienced by YCs may be invisible, especially if YCs do not identify themselves as carers or if their family members do not identify them as YCs, if the professionals are not aware of them, or if YCs do not want to be identified as such [25]. For this reason, it is important that professionals who are or might be in contact with YCs are informed and able to recognize them [26,27,28]. Furthermore, in order to identify and effectively support YCs, their needs and voices need to be heard [23,29] and professionals need to know how they can support them appropriately [28]. Since every YC might have different needs and preferences, it is essential to ask for their perspectives on the type of support they want [29,30].

Regarding its national awareness and policy responses to the situation of YCs, Switzerland finds itself at a ‘preliminary’ level [31]. This means that in Switzerland, there is still little recognition and awareness of YCs, limited but growing research on the topic, no legislation specifically addressing the rights of YCs, and a few dedicated services and interventions [31]. From a nationwide quantitative study has emerged a national prevalence of 7.9% YCs among children and adolescents aged 10–15 [32]. The prevalence in other European countries varies depending on the country of reference and methods used, but in general, it is similar to the one found in Switzerland and is approximately 7–8% [3,33]. Another recent study conducted in Switzerland [28] showed that there is a rather low level of the awareness of YCs among professionals in education, health care and social services and that only a minority (30.3%) of professionals perceived the issue of YCs as being relevant in their occupational context. However, in Switzerland, there is now an increased focus on children’s issues. The recognition and support of YCs are reliant upon ‘non-specific legislation’ or policies related to education, health and social care, children, youth and families, or carers in general (e.g., the program ‘Entlastungsangebote für betreuende Angehörige 2017–2020’ of the Federal Office of Public health, a research program also including minors carers) [24,34]. However, there are no formal assessment tools to identify YCs in Switzerland as yet, which can be found, for example, in the UK [27]. Information is therefore based on a voluntary level for YCs themselves as well as for professionals in the health, education, and social sectors and relies on their awareness of YCs.

Previous studies in Switzerland have mainly focused on YCs under the age of 18 [25,32] or young adult carers between 18 and 25 years old [25]. However, little attention has been paid to the specific group of YCs aged 15–17, the so-called ‘adolescent young carers’ (AYC). AYCs find themselves in a transitionary phase between childhood and adulthood [2,3]. This phase can be considered particularly delicate because several important decisions and changes (e.g., educational pathways) usually take place, often in a relatively short period of time. The outcomes of these decisions can strongly influence the development and professional future including the employment prospects of the young person [2,35]. 

The present paper will focus on AYCs and is based on data collected in the framework of the project ‘Psychosocial support for promoting mental health and well-being among adolescent young carers (AYCs) in Europe’ (ME-WE), funded by the European Union under the Horizon 2020 program [2,3]. The aim of this first cross-national project addressing AYCs was to strengthen their resilience and improve their mental health and well-being. The results presented in this paper are part of the first work package of the ME-WE study, which included an online survey aimed at providing insights into the characteristics, needs and preferences of AYCs in six European countries (the Netherlands, Sweden, Italy, Slovenia, Switzerland, and the UK) and conducted under the lead of the University of Sussex (UK). The first cross-national results have recently been published elsewhere [2], showing that AYCs, on average, tend to have lower health-related quality of life (HRQL) than non-AYCs. More specifically, the findings have shown that there is a statistically significant difference between the HRQL, measured through the mean values of KIDSCREEN 10, among AYCs and respondents without a caring role, with AYCs reporting lower mean scores than their counterparts [2]. Furthermore, among the AYCs across all six countries, 18.3% reported self-harm thoughts because of their caring role; 4.9% reported having thoughts of hurting someone else; 16.1% reported having been bullied, teased, or made fun of at school because of caring; and 33.9% of respondents reported mental health issues related to their caring role [2]. 

The present paper aimed at deepening these results for AYCs in Switzerland and discussing the findings in relation to the Swiss context (e.g., the visibility of AYCs in Switzerland, the awareness of professionals and the services available). In particular, it addressed the following research questions: (1) Which characteristics of AYCs are associated with lower HRQL and with higher levels of mental health problems? (2) Do AYCs who are less visible and who feel less supported report a lower HRQL and more mental health issues than other AYCs?

## 2. Materials and Methods

### 2.1. Participants and Procedures

Data collection took place between April to December 2018 in all six countries (the Netherlands, Sweden, Italy, Slovenia, Switzerland, and the UK) and between January to July 2019 solely in Switzerland, Sweden, the Netherlands, and the UK. The second data collection period was necessary because in the first period, too few a number of AYCs were recruited. In Switzerland, the recruitment strategy took place through vocational training schools and high schools in the German-speaking part of Switzerland. Contact with the school directors was by email and telephone. The schools that agreed to take part in the study received a link with the study information, informed consent form, and the online survey. Participating schools were invited to use a school lesson to give their pupils the chance to take part in the study. Alternatively, they sent the link with the study information, informed consent form and the online survey to the young people via email or the Intranet. In order to avoid the identification of AYCs by their classmates (e.g., because they needed more time to fill in the survey), the survey was designed in such a manner that children who were not AYCs would have taken the same amount of time as AYCs to complete the survey. Non-AYCs were not screened out from the study and their data were used to make a comparison between their HRQL and that of the AYCs. To achieve the targeted sample size of 200 AYCs, the recruitment strategy was slightly adjusted so that additional schools and a few teaching hospitals offering vocational training were contacted between January and July 2019. In total, eleven schools and two hospitals participated in the study.

### 2.2. Measures

The online survey was composed of a demographic section with questions about the characteristics of the young people and questions about the demographic characteristics of the cared for person including the type of health-related condition such as physical, mental, or cognitive health-related condition, addiction, others. The identification of AYCs was based on the following five questions: (1) Do you have someone in your family or someone among your friends or acquaintances with a health-related condition? (2) What type of health-related condition does that person(s) have? (3) Who are these persons? (4) Do you live with the family member(s) or the close friend who has a health-related condition? (5) Do you look after, help, or support any of these family members or close friends with a health-related condition? Only respondents aged between 15 and 17 years old who reported to look after, help or support someone—either for a family member or close friend or both—were classified as AYCs. 

In order to evaluate the HRQL of the participants, the KIDSCREEN 10 questionnaire was used [36,37]. The KIDSCREEN 10 questionnaire includes 10 items aiming at measuring children and young people’s self-reported HRQL. The instrument has been validated in over 13 European countries. A total sum score ranges from 10 to 50 and higher scores indicate greater well-being [37]. As indicators of the risk of poor mental health, young people were also asked if, because of their caring role, they experienced self-harm thoughts, thoughts of harming someone, bullying, and self-reported mental health issues. In addition, in order to measure their visibility, AYCs were asked if (1) their school, (2) their employer, and/or (3) a friend or a close person knew about their caring situation. AYCs were also asked if they had received any kind of the following support: (1) Financial support for themselves and their family; (2) support for their family (e.g., from health professionals, social workers, or from other family members); and (3) support for the AYCs themselves (e.g., from health professionals, social workers, school or close persons). Since the visibility and the types of support received were not mutually exclusive (i.e., the AYCs might have received more than one type of support), the respondents were enabled to indicate more than one possible answer. 

As a first step, the survey was developed in English for all countries and then translated into German for Switzerland. The online survey was hosted on the 1 ka online platform and the data were stored on a server at the University of Ljubljana. After data collection, the data were cleaned, encrypted, and transferred first to the University of Sussex and then to the other project partners for country-specific analyses.

### 2.3. Data Analysis 

Data were analyzed using IBM SPSS Statistics (version 25.0, Armonk, NY, USA: IBM Corp.). Descriptive statistics including the frequencies, means, and standard deviations were computed in order to describe the respondents’ characteristics. Analysis of variance (ANOVA) and chi-square tests were used to compare the mean scores and proportions, respectively, across the groups. For age group and type of settlement, χ^2^ tests with post-hoc z-scores and Bonferroni correction were used. Interpretation of the results was based on statistical significance (*p* < 0.05) and effect size. A Cohen’s *d* of 0.20 was interpreted as small, 0.50 medium, and 0.80 large. A Cramer’s *V* of 0.10 was considered small, 0.30 medium, and 0.50 large [38]. For the analyses, the categories where the numbers were very small (i.e., transgender or others for gender) were not included. 

### 2.4. Ethics

For data collection in Switzerland, a clarification of responsibility was submitted and secured by the Cantonal Ethics Committee of the Canton of Zurich in March 2018. Before filling out the questionnaire, AYCs were requested to read the study information and to sign an informed consent form. The study information and consent form were easy to understand and worded age-appropriately. They included the foreseen benefits and possible risks of participation in the study as well as the information that they were able to withdraw their participation at any time without any negative consequence. The General Data Protection Regulations (GDPR) were followed in addition to the national laws and EU laws on data protection. The participants’ anonymity was guaranteed in all phases of the study.

## 3. Results

### 3.1. Characteristics of the Participants

Overall, 2343 young people living in the German-speaking part of Switzerland participated in the study. Of the 871 aged 15–17 years, 240 (27.6%) were identified as AYCs. The sociodemographic characteristics of the sample (15–17 year old AYCs and non-carers) are presented in Table 1. The great majority of AYCs (80.8%) were young women, whilst 62.7% of the 15–17 year old non-carers were female. Furthermore, the majority of respondents were 17 years old (58.3% of AYCs, 57.2% of non-carers), lived in the countryside (56.6% vs. 43.8%), had Swiss nationality (75% vs. 73.7%), and lived with their mother (93.8% vs. 95.7%), with their father (68.3% vs. 81.8%), and/or with their siblings (63.7% vs. 57.2%). 

Among the AYCs, 75.8% reported having someone in the family with a health-related issue and 66.7% reported having a close person who did not belong to the family and who was suffering from a health-related issue. Among the ones having a family member with a health-related issue, 37.9% reported that this person had a physical disability, 35.8% a mental health issue, 19.2% a cognitive disability (e.g., dementia, autism or Down syndrome) and 11.7% addiction issues. A few AYCs (9.2%) reported having someone in the family with another health-related issue not listed in the survey. When investigating the type of relationship AYCs had with the cared for person, 22.9% of the AYCs reported that this person was their mother, 12.1% their father, 22.1% a sibling, 39.6% a grandparent, and 19.6% reported that this person was another family member (e.g., a cousin, aunt or uncle, or the partner of their mother or father). Among the AYCs caring for a family member, 43.3% reported living with this person. 

A total of 160 AYCs (66.7%) reported having a person close to them, but who was not a family member, who was suffering from a health-related issue. In 22.1% of the cases, this person had a physical disability, 42.9% of cases had a mental health issue, 8.8% of them had a cognitive disability, and 15.4% addiction issues. In most of the cases, this person was a friend (39.6%), followed by a colleague (10%), a neighbor (5%), and by a partner (5%). In 26.2% of the cases, this person was someone else (e.g., an ex-boyfriend or girlfriend). A few AYCs (2.9%) reported living with this person. 

When asked how long they had been caring for their family member or someone close to them, 20% of AYCs reported caring for this person as long as they could remember. The majority of other respondents had also been caring for someone close to them for a long time: 1.2% of respondents reported caring for this person for 1 or 2 years, 3.3% between 3 and 5 years, 14.7% between 6 and 10 years. The majority of AYCs reported that they had been caring for someone for between 11 and 15 years (40.5%) and between 16 and 17 years (11.6%). Since the respondents were aged between 15 and 17 years old, this means that these AYCs started their caring roles at a very early age (e.g., by providing emotional support). In some cases, this can mean that they were even born in a situation of physical, mental, or cognitive illness or disability in the family.

### 3.2. Characteristics of AYCs Associated with HRQL and Mental Health

The mean value of KIDSCREEN 10 among the AYCs was 33.29 (SD = 7.12). Table 2 reports the mean values with standard deviations of KIDSCREEN 10 among the AYCs in different sociodemographic groups, based on their age, gender, type of settlement, and nationality. No significant group difference based on age group (*F*(2, 221) = 0.774, *p* = 0.462), gender (*F*(1, 220) = 0.004, *p* = 0.949), type of settlement where the AYCs lived (*F*(4, 126) = 0.932, *p* = 0.448), or having Swiss nationality (*F*(1, 222) = 2.394, *p* = 0.123) was found. 

When testing the associations between the sociodemographic characteristics and indicators for mental health risk (self-harm thoughts, thoughts of harming someone, being bullied, and reported mental health issues), small but significant associations of gender (*χ*^2^ (1) = 6.151, *p* = 0.046, Cramer’s *V* = 0.177), and having Swiss nationality (*χ*^2^ (1) = 3.966, *p* = 0.046, Cramer’s *V* = 0.142) with self-reported mental health issues were found. A significantly higher proportion of female than male AYCs and a significantly higher proportion of AYCs with rather than without Swiss nationality reported mental health issues. All other associations were nonsignificant. The proportions are presented in Table 3. 

### 3.3. Support and Visibility of AYCs, HRQL, and Mental Health

A minority of AYCs reported that they or their family received some kind of support: 21.7% (*n* = 52) said that their family was receiving financial support, 17.1% (*n* = 41) stated that the family received support from professionals or other family members, and 15.4% (*n* = 37) declared that they themselves as AYCs were receiving support (e.g., by homecare services, health professionals, or in the form of counseling or information). A few AYCs (8.3%, *n* = 20) reported that their school knew about their home situation, 12.1% (*n* = 29) that their employers knew about their caring role, and 58.3% (*n* = 137) reported that a friend or a close person was informed about their home situation. 

Table 4 reports the mean KIDSCREEN 10 values among the AYCs, according to the type of support received. A significant but modest effect (*d* = 0.24) between support for the AYCs and HRQL was found, with AYCs who received support for themselves reporting a higher HRQL than the AYCs who did not receive this kind of support (*F* (1, 131) = 3.247, *p* = 0.041). Furthermore, significant but modest effects were found between the school (*d* = 0.44) and/or the employer (*d* = 0.40) knowing about the caring situation and the HRQL. Indeed, AYCs who reported that their school knew about their caring situation reported higher HRQL than the AYCs whose school was not informed (*F*(1, 131) = 4.364, *p* = 0.039). AYCs whose employers (*F*(1, 131) = 4.573, *p* = 0.034) knew about the situation at home also reported higher mean scores than their counterparts. No significant differences in the KIDSCREEN 10 mean scores were found between the AYCs who received financial support and AYCs who were not supported that way, nor between the AYCs who reported that their family received support and AYCs who did not receive support addressing their family. 

The proportions of AYCs with self-harm thoughts, thoughts of harming someone, being bullied, and reported mental health issues by type of support or visibility received were calculated and compared using the χ^2^ test. A small and almost significant association (Camer’s *V* = 0.166, χ^2^ (1) = 3.728, *p* = 0.05) was found between the school knowing about the caring situation and self-reported mental health issues. The proportion of AYCs reporting that the school did not know about their caring situation was higher among AYCs who reported mental health issues. All other proportions, which were not significant, are reported in Table 5 and Table 6. 

## 4. Discussion

The study aim was to gain insights into the well-being and mental health of AYCs in Switzerland and to identify characteristics associated with poor HRQL and perceived mental ill health. Furthermore, it aimed to investigate whether the visibility of AYCs and the support they received were associated with their HRQL and mental health issues. The results of the ME-WE study including the ones presented in this paper, suggest a high risk of mental distress and an urgent need for tailored and appropriate psychosocial support for AYCs as well as an urgent need to increase their visibility [2,3,39].

### 4.1. Most Important Findings

In the present study among AYCs, 75.8% reported having someone in the family with a health-related issue and 66.7% reported having a close person who did not belong to the family and who was suffering from a health-related issue.

When considering the reported mental health issues related to the caring role, the results showed that female AYCs reported mental health issues more frequently than male AYCs. This result confirms what has been found in previous studies, suggesting that female young carers have a higher risk of mental distress than their male counterparts [40,41,42]. Furthermore, although the effect size was very small, the findings of the present study suggest that AYCs with Swiss nationality reported more mental health issues than AYCs with nationalities other than Swiss. However, according to the literature, migrant YCs and AYCs have a higher risk of mental health issues as their caring role often co-exists within their experience of trauma, displacement, and instability [2,43,44]. The present study did not consist of sufficient data to further investigate the possible associations between the migratory background of AYCs and poor mental health, for example, because we did not include more specific questions concerning the immigration experience lived by the AYCs and their families or their immigration status. Additionally, further aspects should be taken into consideration when investigating the effects of a migratory background on the mental health consequences of a caring role. For example, there might be cultural differences in the perception and disclosure of mental health or a different access to health services. Therefore, despite the modest effect, this finding may not be meaningful and further research on this topic is needed. 

The data in the present study show that there is a relation between HRQL and the support received by AYCs and their visibility, namely, if their school or employer knew about the caring situation. In other words, AYCs reporting that their school or employer knew that they had a caring role also reported higher KIDSCREEN10 scores. Furthermore, an association was found between the self-reported mental health issues and the visibility received by AYCs through their school knowing about the caring situation. The proportions of AYCs reporting negative mental health issues were significantly lower when the school was informed about the caring situation. 

### 4.2. Visibility Is a Key Dimension to Better HRQL and Mental Health

Being an AYC is still invisible or hidden in Switzerland. The findings of the present study showed that only 8.3% of the surveyed AYCs reported that their school was informed about the caring role and in 12.1% of the cases, their employers were informed. A little more than a half had a friend or a close person who was informed about their caring role. When comparing these results with those of the other countries, it shows that the UK, for example, has higher percentages: 59% of the AYCs reported awareness by the school and 67% by a close friend [2]. This is even more relevant when considering that AYCs participating in the present study who reported that their school and/or employer was informed about their situation also reported higher KIDSCREEN10 scores and fewer self-reported mental health issues when their school was informed.

### 4.3. AYCs Need Tailored Support

A few AYCs reported that their families were receiving some kind of support and even less were receiving support for themselves. Therefore, it seems that only a few AYCs received support targeting themselves as AYCs and that was tailored to the specific needs that can arise from the caring role. However, this type of support resulted in being particularly relevant, as AYCs receiving support for themselves reported higher KIDSCREEN 10 mean scores. Financial support or support targeting their families was not significantly associated with the AYCs’ HRQL nor to their mental health. Therefore, the findings of the present study suggest that dedicated support that targets the AYCs themselves may be more beneficial than other types of support. However, in Switzerland, there are only a few services and programs aiming at offering tailored support to AYCs, and most of these programs are offered at a local or regional level and do not cover all the language regions yet [23,31,45]. The finding that financial support is not associated with the AYCs’ HRQL nor to their mental health also suggests that AYCs might prefer in-kind services over cash services. However, it should be questioned whether more support targeting the AYCs is always a possible solution, especially when levels of caring are inappropriate and young people are not able to attain their rights (e.g., their right to education) [23,24]. In these cases, appropriate caring delivered by formal health care services and professionals to the person in need, so that young people are not forced to take over a caring role, might benefit the AYCs’ mental health more than support targeting the AYCs themselves. 

The findings of the present study have also shown that AYCs reported higher KIDSCREEN 10 mean scores and fewer self-reported mental health issues if their school and/or employer knew about the caring situation. This means that the visibility of AYCs seems to be positively associated with their HRQL and their mental health. However, previous studies have shown that the awareness and visibility of AYCs in Switzerland are quite low [23,28,31,45]. More efforts should therefore be put in place in order to increase the visibility of AYCs, and consequently to support their HRQL and mental health. 

### 4.4. AYCs’ Mental Health and Support during the COVID-19 Pandemic 

The data in the present study were collected before the COVID-19 pandemic and they therefore do not represent the AYCs’ mental health, the visibility, nor the support received during the pandemic. However, other studies reported that during the COVID-19 pandemic, the number of AYCs experiencing low HRQL and mental health issues is likely to have increased. Indeed, the COVID-19 pandemic and the protective measures have increased the stress and pressure of young people and of younger carers in particular [46]. Due to lockdowns, formal and informal care services for the cared for person became difficult or were even disrupted, and therefore the YCs had to shoulder a greater proportion of caring tasks. Furthermore, the school closures led to fewer contacts with peers and more social isolation as well a further impairment in their education [46]. YCs were also worried about the health of the person they cared for including the fear of COVID-19 transmission. This increase in caring roles, anxieties, and isolation led to a higher risk of negative mental health [47]. Given the likelihood of future pandemics, it is even more important to study the mental health and well-being of AYCs as well as how they can best be supported, even in unusual situations such as during the COVID-19 pandemic, where the support targeting the AYCs might have been disrupted. 

### 4.5. Strengths and Limitations

ME-WE is the first international study investigating the mental health and well-being of the subgroup of AYCs aged 15–17 years—in a vulnerable transitionary phase—in Europe and to substantially generate new knowledge about AYCs [48]. Aside from the significance of the international study and the cross-country comparisons, country-specific results are important in order to address Swiss national policy and support services for AYCs. The present study highlights, for the first time, data about the HRQL, perceived mental health, and support received by AYCs in Switzerland, therefore contributing to the understanding of the need for the support of AYCs. A third strength of the present study was the use of a widely used, validated psychometric instrument (i.e., KIDSCREEN 10) combined with additional questions to investigate the mental feelings of the respondent AYCs, which allowed the research team to gain basic information about the AYCs’ well-being and mental health as well as their exacerbations.

The present study had some limitations. First, the recruitment of participants took place in vocational training schools, particularly in the health sector, teaching hospitals with vocational training, and high schools in the German-speaking part of Switzerland. The high proportion of vocational schools for health professions recruited for the present study may explain the inequality of gender (female 80.8% versus male 18.8%). Indeed, vocational training for health professions in Switzerland are mainly attended by young women [49]. This limits the generalizability of the findings. Furthermore, AYCs pursuing an education in the health sector might recognize their role differently to AYCs not in health education or might be more aware of the needs of the cared for person as well as the existing support offers. The recruitment through post-compulsory schools also posed the limitation that AYCs who had not had the chance of pursuing a further education were not included in the sample. Such AYCs might have experienced different or greater consequences of the caring role. Further research is therefore needed on the challenges experienced by AYCs not engaged in further education. Second, the number of AYCs participating in the survey was small, especially when analyzing some specific sociodemographic characteristics. Because some characteristics (e.g., living in a big city) were under-represented in this study, future research involving larger and more diverse samples of AYCs is warranted to replicate our findings. Furthermore, some questions of the KIDSCREEN 10, questions investigating the mental health of AYCs or the support received were not answered by all respondents, leading to missing values and reducing the number of AYCs considered for the analysis. The missing values might be due to the fact that the questions were not mandatory. The reason was that the respondents participated on a voluntary basis, and therefore, we decided not to force them to answer any questions that might have been uncomfortable for them. Third, the use of self-report measures and the setting of data collection (i.e., the classroom) might have introduced social desirability biases. Indeed, for example, mental or addictive illnesses, which are often stigmatized [50,51], might have contributed to the underreporting. The same may be true for the questions regarding self-harm thoughts and thoughts of harming someone else. AYCs were also identified through a self-declared measure by asking them if they supported someone close to them or a family member because of an illness or disability. Since many AYCs do not recognize themselves as such, this measure, based on a broad definition of ‘caring’ (i.e., ‘to look after, help or support’), was deemed as appropriate. However, the experiences of AYCs can be very heterogeneous, and this question did not provide information concerning the type of tasks provided by AYCs. Furthermore, the present paper did not provide information on the intensity of the caring tasks, and the results on the intensity of the caring tasks are reported elsewhere [2]. Finally, a more in-depth mental health risk assessment was not conducted, using, for example, depression and suicide screening adapted for adolescents and young people (e.g., [52]). This topic warrants further investigation. 

### 4.6. Recommendations for Further Research, Policy, and Practice

The present study sheds light on the current situation of AYCs in Switzerland. This is vital, particularly in this transitionary and delicate phase between childhood and adulthood, where important decisions and changes (e.g., educational pathways) usually take place. Further research is needed on the impact of caring roles on the mental health and well-being of AYCs and YCs including the impact that the caring role might have on the future lives of AYCs. Furthermore, it should be further investigated how resilience can be best promoted and how YCs and AYCs can be effectively supported. The findings presented in this paper do provide an overview of the relation between HRQL and the visibility and support received. However, the need for support might vary depending on the specific caring situation. Further research is needed to understand what difficulties are reported by AYCs experiencing different caring situations (e.g., caring for a family member or for someone outside the family, living in the same household of the cared for person or not) and what kind of support is needed depending on the situation. 

More efforts at the policy and practice level in order to support AYCs and to increase their visibility are also necessary. In Switzerland, there are no specific laws and only a few policy initiatives recognizing and supporting AYCs [24,31] as well as few support offers [23]. Although the awareness and visibility of YCs and AYCs has slightly increased in the past few years and some support offers have emerged, thus allowing Switzerland to move from an emerging to a preliminary level on the classification of in-country awareness and policy responses to young carers of Leu and Becker [31,53], policy and support offers for AYCs as well as their visibility is still limited. There are some promising support offers such as informal meetings for young carer ‘get-togethers’ [54], but these take place mostly at a local or regional level. Further efforts and more cooperation between practice organizations would allow for such initiatives to expand in other linguistic regions and other cantons of Switzerland, reaching AYCs who are, at the moment, not able to benefit from such offers because their region is not yet served. Further cooperation with policy actors, schools, health, and social organizations as well as interprofessional cooperation within health, educational, and social professionals [55] would also benefit AYCs by increasing their visibility. In particular, the findings of the present study confirm that raising the awareness and the visibility of AYCs in the educational sector is important. This might be reached, for example, with massive awareness campaigns and educational training for school staff in Switzerland, in order to make teachers, school social workers, and educational professionals able to identify the YCs in general and to guide them to the appropriate support. In an ideal world, there should be no ‘wrong doors’ for young carers, who should be identified and successfully supported regardless of which organizations or professionals are contacted first [56].

## 5. Conclusions

The present manuscript addressed the question of which characteristics of AYCs are associated with lower HRQL and with a higher level of mental health problems as well as whether AYCs who felt less supported or were less visible reported lower HRQL and more negative mental health issues than AYCs who received sufficient support. The findings of the present study show that AYCs in Switzerland who reported a lower HRQL and more mental health issues also less often reported receiving support for themselves or that their school, employer or a friend knew about the caring situation. These results show how important and supportive it is for AYCs when they are visible to their school, employers, or a friend. However, quoting J. W. Goethe, ‘we only see what we know’ [57], so knowing and seeing are connected. Therefore, it is important to increase the visibility of AYCs by raising the awareness of professionals who are in contact with these young people, particularly those in the educational sector. More efforts should be taken in order to make this invisible group of vulnerable children visible to professionals and to society as a whole. 

## Figures and Tables

**Table 1 ijerph-20-03963-t001:** The sociodemographic characteristics of the sample.

	AYCs (*n* = 240) % (*n*)	Non-Carers (*n* = 631) % (*n*)
Gender		
Female	80.8% (193)	62.7% (391)
Male	18.8% (45)	36.1% (225)
Transgender	0% (0)	0.3% (2)
Other	0.4% (1)	0.6% (4)
Age		
15	8.3% (20)	7% (44)
16	33.3% (80)	35.8% (226)
17	58.3% (140)	57.2% (361)
Type of settlement		
Big city	5.1% (7)	6.8% (30)
Suburb of a big city	11% (15)	14.4% (64)
Small city	24.3% (33)	31.8% (141)
Village in the countryside	56.6% (77)	43.8% (194)
Farm/house in the countryside	2.9% (4)	3.2% (14)
Nationality		
Swiss	75% (180)	73.7% (465)
German	10% (24)	7.4% (47)
Italian	11.7% (28)	9.8% (62)
French	1.3% (3)	1.6% (10)
Portuguese	1.7% (4)	1.6% (10)
Kosovan	4.6% (11)	7.8% (49)
Spanish	2.9% (7)	1.6% (10)
Serbian	2.9% (7)	2.9% (18)
Turkish	1.7% (4)	4% (25)
Other	24.2% (58)	21.7% (137)
Whom they live with		
Mother	93.8% (225)	95.7% (604)
Father	68.3% (164)	81.8% (516)
New partner of the mother/father	7.1% (17)	7.4% (47)
Siblings	63.7% (153)	57.2% (361)
Grandmother	3.8% (9)	3.6% (23)
Grandfather	2.5% (6)	2.5% (16)
Others	5.1% (12)	4.6% (29)

(*n*) Unweighted number of cases. The total can differ from *n* = 240 or *n* = 631 due to missing values. Percentages do not always add up to exactly 100% because of missing values. For nationality and persons the AYCs lived with, the percentages added up to more than 100% because multiple answers were possible.

**Table 2 ijerph-20-03963-t002:** Comparison of the mean KIDSCREEN 10 scores (SD) of the AYCs based on age, gender, type of settlement, and nationality.

Characteristics of AYCs	KIDSCREEN 10 Score (Mean (SD))
Age (*n* = 224)	
15 (*n* = 20)	35.15 (8.74)
16 (*n* = 74)	33.24 (7.31)
17 (*n* = 130)	33.02 (6.74)
Gender (*n* = 222)	
Female (*n* = 179)	33.05 (7.09)
Male (*n* = 43)	34.33 (7.19)
Type of settlement (*n* = 128)	
Big city (*n* = 7)	29.29 (9.79)
Suburb of a big city (*n* = 15)	31.60 (6.97)
Small city (*n* = 33)	32.06 (5.99)
Village in the countryside (*n* = 73)	33.79 (7.81)
Farm/house in the countryside (*n* = 3)	34.00 (3.61)
Swiss nationality (*n* = 224)	
Yes (*n* = 169)	33.70 (6.81)
No (*n* = 55)	32.00 (7.91)

Note: The total can differ from *n* = 240 due to missing values.

**Table 3 ijerph-20-03963-t003:** Self-harm thoughts, thoughts of harming someone, being bullied, and the reported mental health issues by the demographics.

Demographics of AYCs	Self-Harm Thoughts		Thoughts of Harming Someone		Being Bullied		Reported Mental Health Issues	
	Yes	No	Yes	No	Yes	No	Yes	No
	% (*n*)	% (*n*)	% (*n*)	% (*n*)	% (*n*)	% (*n*)	% (*n*)	% (*n*)
Age (*n* = 240)								
15 (*n* = 20)	9.8% (4)	7.6% (12)	10% (1)	8% (15)	5.7% (2)	8.6% (14)	7.5% (4)	8.3% (12)
16 (*n* = 80)	34.1% (14)	32.9% (52)	40% (4)	33.2% (63)	34.3% (12)	33.3% (54)	34% (18)	33.3% (48)
17 (*n* = 140)	56.1% (23)	59.5% (94)	50% (6)	58.8% (110)	60% (21)	58% (94)	58.5% (31)	58.3% (84)
Gender (*n* = 238)								
Female (*n* = 193)	80% (32)	81% (128)	70% (7)	81.7% (152)	88.6% (31)	78.9% (127)	92.5% (49) *	76.9% (110) *
Male (*n* = 45)	20% (8)	18.4% (29)	30% (4)	17.7% (33)	11.4% (4)	20.5% (33)	7.5% (4) *	22.4% (32) *
Type of settlement (*n* = 132)								
Big city (*n* = 7)	0% (0)	6.5% (6)	14.3% (1)	4.5% (5)	8.7% (2)	4.2% (4)	4% (1)	5.4% (5)
Suburb of a big city (*n* = 15)	11.5% (3)	8.6% (8)	0% (0)	9.1% (10)	17.4% (4)	7.4% (7)	8% (2)	8.7% (8)
Small city (*n* = 33)	23.1% (6)	22.6% (21)	57.1% (4)	30.9% (23)	21.7% (5)	23.2% (22)	32% (8)	19.6% (18)
Village in the countryside (*n* = 77)	65.4% (17)	59.1% (55)	28.6% (2)	62.7% (69)	43.5% (10)	64.2% (61)	56% (14)	63% (58)
Farm/house in the countryside (*n* = 4)	0% (0)	3.2% (3)	0% (0)	2.7% (3)	8.7% (2)	1.1% (1)	0% (0)	3.3% (3)
Swiss nationality (*n* = 240)								
Yes (*n* = 180)	73.2% (30)	78.5% (124)	60% (6)	78.6% (147)	74.3% (26)	79% (128)	67.9% (36) *	81.3% (117) *
No (*n* = 60)	26.8% (11)	21.5% (34)	40% (4)	21.4% (40)	25.7% (9)	21% (34)	32.1% (17) *	18.8% (27) *

Note: The total can differ from *n* = 240 due to missing values. * *p* < 0.05.

**Table 4 ijerph-20-03963-t004:** Comparisons in the mean KIDSCREEN 10 score (SD) based on receiving support.

KIDSCREEN 10 Score and…	Mean Score (SD)
Financial support (*n* = 211)	
Yes (*n* = 52)	32.35 (8.10)
No (*n* = 159)	33.51 (6.78)
Support addressing the family (*n* = 212)	
Yes (*n* = 40)	32.80 (7.75)
No (*n* = 172)	33.36 (7.02)
Support addressing the AYC (*n* = 178)	
Yes (*n* = 36)	34.08 (8.65) *
No (*n* = 142)	32.42 (6.60) *
School knows about the situation (*n* = 182)	
Yes (*n* = 20)	35.40 (5.48) *
No (*n* = 162)	32.36 (7.02) *
Employer knows about the situation (*n* = 149)	
Yes (*n* = 27)	34.56 (7.73) *
No (*n* = 122)	31.80 (6.69) *
Friend knows the situation (*n* = 186)	
Yes (*n* = 137)	32.85 (7.23)
No (*n* = 49)	32.49 (6.56)

Note: The total can differ from *n* = 240 due to missing values. * *p* < 0.05.

**Table 5 ijerph-20-03963-t005:** Self-harm thoughts, thoughts of harming someone, being bullied, and reported mental health issues by the type of support received.

Support Received and…	Self-Harm Thoughts (*n* = 199)		Thoughts of Harming Someone (*n* = 199)		Being Bullied (*n* = 200)		Reported Mental Health Issues (*n* = 189)	
	Yes (*n* = 41)	No (*n* = 158)	Yes (*n* = 11)	No (*n* = 188)	Yes (*n* = 36)	No (*n* = 164)	Yes (*n* = 64)	No (*n* = 125)
	% (*n*)	% (*n*)	% (*n*)	% (*n*)	% (*n*)	% (*n*)	% (*n*)	% (*n*)
Financial support (*n* = 211)								
Yes (*n* = 52)	32.3% (10)	20.2% (21)	12.5% (1)	23.6% (30)	35.5% (11)	19.2% (20)	30.8% (12)	19.8% (19)
No (*n* = 159)	67.7% (21)	79.8% (83)	87.5% (7)	76.4% (97)	64.5% (20)	80.8% (84)	69.2% (27)	80.2% (77)
Support addressing the family (*n* = 212)								
Yes (*n* = 40)	25.8% (8)	15.4% (16)	12.5% (1)	18.1% (23)	15.6% (5)	18.4% (19)	17.5% (7)	17.9% (17)
No (*n* = 172)	74.2% (23)	84.6% (88)	87.5% (7)	81.9% (104)	84.4% (27)	81.6% (84)	82.5% (33)	82.1% (78)
Support addressing the AYC (*n* = 178)								
Yes (*n* = 36)	12.9% (4)	15.2% (16)	12.5% (1)	14.8% (19)	9.4% (3)	16.3% (17)	15% (6)	14.6% (14)
No (*n* = 142)	87.1% (27)	84.8% (89)	87.5% (7)	85.2% (109)	90.6% (29)	83.7% (87)	85% (34)	85.4% (82)

Note: The total can differ from *n* = 240 due to missing values.

**Table 6 ijerph-20-03963-t006:** Self-harm thoughts, thoughts of harming someone, being bullied, and reported mental health issues by visibility received.

Support Received and…	Self-Harm Thoughts (*n* = 199)		Thoughts of Harming Someone (*n* = 199)		Being Bullied (*n* = 200)		Reported Mental Health Issues (*n* = 189)	
	Yes (*n* = 41)	No (*n* = 158)	Yes (*n* = 11)	No (*n* = 188)	Yes (*n* = 41)	No (*n* = 158)	Yes (*n* = 11)	No (*n* = 188)
	% (*n*)	% (*n*)	% (*n*)	% (*n*)	% (*n*)	% (*n*)	% (*n*)	% (*n*)
School knows about the situation (*n* = 182)								
Yes (*n* = 20)	6.5% (2)	11.4% (12)	12.5% (1)	10.2% (13)	9.4% (3)	10.6% (11)	2.5% (1) *	13.5% (13) *
No (*n* = 162)	93.5% (29)	88.6% (93)	89.8% (115)	89.8% (115)	90.6% (29)	89.4% (93)	97.5% (39) *	86.5% (83) *
Employer knows about the situation (*n* = 149)								
Yes (*n* = 27)	9.7% (3)	21% (22)	12.5% (1)	18.8% (24)	12.5% (4)	20.2% (21)	12.5% (5)	20.8% (20)
No (*n* = 122)	90.3% (28)	79% (83)	87.5% (7)	81.3% (104)	87.5% (28)	79.8% (83)	87.5% (35)	79.2% (76)
Friend knows the situation (*n* = 186)								
Yes (*n* = 137)	67.7% (21)	72.4% (76)	50% (4)	72.7% (93)	78.1% (25)	69.2% (72)	82.5% (33)	66.7% (64)
No (*n* = 49)	32.3% (10)	27.6% (29)	50% (4)	27.3% (35)	21.9% (7)	30.8% (32)	17.5% (7)	33.3% (32)

Note: The total can differ from *n* = 240 due to missing values. * *p* < 0.05.

## Data Availability

Data sufficient for the reader to validate the article results can be made available as appropriate on request to the first author.

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
