# Peer review of "Visibility as a Key Dimension to Better Health-Related Quality of Life and Mental Health: Results of the European Union Funded “ME-WE” Online Survey Study on Adolescent Young Carers in Switzerland"

_ijerph, 2023, doi:10.3390/ijerph20053963_

Round 1

Reviewer 1 Report

Dear Authors,

Thank you for this work on an interesting and really important topic.

Your manuscript is particularly well written, very clear and easy to read, especially the introduction which gives a good understanding of what a young carer is, and the Swiss context. I particularly liked your reflection on how we should look at YCs regarding their role and the manner to perceive / support them (lines 88-93).

I don’t have much remark to make of your work, but a few points caught my attention.

1 – In your survey, the status of YCs is only measured by a self-declaration. Although indicating whether the youth is confronted with the illness of a relative and provides support could be sufficient, it might have been more interesting and relevant to use a validated questionnaire to measure YC’s status. Why didn’t you use one?

Indeed, in your introduction, you mentioned that many YCs didn’t identify themselves as such.  Don’t you think you’ve missed some YCs?

It might be interesting to compare (1) self-reported YCs, (2) youth who reported facing a relative’s illness, and (3) youth who didn’t face a relative’s illness and were not YC, to identify any differences.

2 – The lack of homogeneity according to the variables due to the missing values obstruct a good understanding as well as the reading. Several questions come to my mind:

              . In your opinion, why are there so many missing values?

              . Wasn’t it possible to make the questions mandatory to avoid these missing data?

3 – You said line 393 « Few AYCs reported that their families were receiving some kind of support and even less receiving support for themselves. Therefore, it seems that any support AYCs received is mostly not tailored to their specific needs”. How could you conclude that? The previous sentence does not allow it. Some explanations are missing here.

4 – It would have been interesting here to directly ask the YCs what they thing of the support provided to them (or to their family) and what do they need regarding their role of YC. Did you investigate it elsewhere?

Reviewer 2 Report

Paper Review

This is an important and timely paper exploring the characteristics of adolescent Young Carers in Switzerland, including their support and wellbeing. The paper makes an important contribution to the literature, by highlighting characteristics of this group of young people and to better understand their wellbeing and their needs for support.

In my reading of the manuscript I have identified a few points for the authors to consider.

Review comments:

1. The identification of AYC includes the question about whether the young person supports someone close to them or a family member. It is understandable and necessary for this to be a wide definition. However, it is possible that the experiences of the AYC may be extremely varied, and the intensity of caring duties may be very heterogeneous. While it may not have been be pragmatic to collect detailed information about the nature of caring duties, it might be helpful to recognise this in the paper, and discuss the implications for the findings. If the young people did give an example of the tasks/nature of caring they perform for another person or family member, it would be useful to include this.

2. The majority of AYC (around 66%) mentioned someone close to them who they do not live with (only 2.9% did) rather than a family member. This may shape the nature of the caring experience, as well as the social/emotional burden associated with it. While it does not diminish the mental or physical load the caring duty may carry, it is possible that the nature of the experience is qualitatively different for those caring for their own parent/sibling (and therefore perhaps missing out on age typical support with every day life) vs those caring for someone else, but still experiencing age appropriate support from within their own family. Young carers whose own parent is unwell with a mental health difficulty and can therefore not adequately look after themselves or their dependents, have likely a different experience to those who are looked after adequately by their own parents/family, but have a caring responsibility to someone else. The social-emotional burden is likely greater for those who don’t benefit from within family support, because it is this close family who they are caring for. It is possible, therefore, that the findings in this paper underestimate the difficulties AYC may face, as the sample includes more than half who care for someone outside the family, and therefore are likely to have access to within family support system.  Perhaps this is something the authors could comment on. 

3. The authors acknowledge that much of the recruitment took place in healthcare training settings and that this may restrict the demographic (e.g. predominantly female respondents). It is possible that there is a further consideration about the potential bias of this demographic. YC seem to be more likely to pursue caring roles for future job and careers than non-YC peers, and it is also possible that they might be more able or willing to recognise their YC identity than those not immersed in professional care training. It would be useful for the authors to consider how this might impact the interpretation of findings from other demographics of young carers (are these perhaps the most likely to be identified?).

4. The finding relating to greater health related concerns within AYC with Swiss nationality is, as the authors acknowledge, unexpected. It is possible that this result is purely coincidental, or spurious, especially as the effect size is fairly small. However, is it possible that there is some relationship to an ethnic or cultural difference in the discourse around disclosing mental ill health, or something related to the health insurance/medical access of Swiss nationals vs other nationals? In any case, the explanation as it stands feels incomplete. It may be best to state the unexpected direction of this finding and that despite the modest effect, it may not be meaningful.

5. YC often experience restricted life chances, e.g. at further education (post 16years). Perhaps the YCs attending further education/vocational college are faring better than those who were unable to go to further education. Again if this is a possibility, this might result in the findings showing an under-representation of the potential challenges faced by YC, and it may be useful to acknowledge and comment on this in discussing the findings.

Some minor comments

A general suggestion for the Reporting of %: Please avoid starting a sentence with the numerical percentage, this breaks the flow of the narrative and can make it more difficult to read and follow.

Page 7, line 272 onwards. The description is somewhat confusing as there is a statistic suggesting that 40.5% of respondents had been caring for 11-15 years. The age of respondents is 15-17 however, which would suggest that some are only 2 at the time they began their caring duty (and likely before this is possible). It might help to explain this finding.

Page 10 is the type of support received mutually exclusive, or could some individuals have received more than one of the support types mentioned? It could be helpful to clarify this.

Results, page 10 line 313 onwards, remove “slightly”. It would be better to say the effects were modest, e.g. “Significant but modest effect” as it is not typical to describe results as being “slightly” significant.  

Page 14 line 352 should read they not their
